# *Lithospermum erythrorhizon* Alleviates Atopic Dermatitis-like Skin Lesions by Restoring Immune Balance and Skin Barrier Function in 2.4-Dinitrochlorobenzene-Induced NC/Nga Mice

**DOI:** 10.3390/nu13093209

**Published:** 2021-09-15

**Authors:** Jin-Su Oh, Sang-Jun Lee, Se-Young Choung

**Affiliations:** 1Department of Life and Nanopharmaceutical Sciences, Graduate School, Kyung Hee University, 26, Kyungheedae-ro, Dongdaemun-gu, Seoul 02447, Korea; ok9638@naver.com; 2Holistic Bio Co., Ltd., Seongnam 13494, Korea; leesjun2006@gmail.com; 3Department of Preventive Pharmacy and Toxicology, College of Pharmacy, Kyung Hee University, 26, Kyungheedae-ro, Dongdaemun-gu, Seoul 02447, Korea

**Keywords:** atopic dermatitis, *Lithospermum erythrorhizon*, NC/Nga, Th1, Th2, Th17, Th22, immune balance, skin barrier function

## Abstract

The incidence of atopic dermatitis (AD), a disease characterized by an abnormal immune balance and skin barrier function, has increased rapidly in developed countries. This study investigated the anti-atopic effect of *Lithospermum erythrorhizon* (LE) using NC/Nga mice induced by 2,4-dinitrochlorobenzene. LE reduced AD clinical symptoms, including inflammatory cell infiltration, epidermal thickness, ear thickness, and scratching behavior, in the mice. Additionally, LE reduced serum IgE and histamine levels, and restored the T helper (Th) 1/Th2 immune balance through regulation of the IgG1/IgG2a ratio. LE also reduced the levels of AD-related cytokines and chemokines, including interleukin (IL)-1β, IL-4, IL-6, tumor necrosis factor-α (TNF-α), thymic stromal lymphopoietin, thymus and activation-regulated chemokine, macrophage-derived chemokine, regulated on activation, normal T cell expressed and secreted, and monocyte chemoattractant protein-1 in the serum. Moreover, LE modulated AD-related cytokines and chemokines expressed and secreted by Th1, Th2, Th17, and Th22 cells in the dorsal skin and splenocytes. Furthermore, LE restored skin barrier function by increasing pro-filaggrin gene expression and levels of skin barrier-related proteins filaggrin, involucrin, loricrin, occludin, and zonula occludens-1. These results suggest that LE is a potential therapeutic agent that can alleviate AD by modulating Th1/Th2 immune balance and restoring skin barrier function.

## 1. Introduction

Atopic dermatitis (AD) is most common in early childhood, however it is also present in adults [1]. AD is accompanied by persistent itching and clinical symptoms such as eczema, erythema, dryness, lichenification, and abrasions [2,3,4,5,6]. AD is an early stage of the “atopic march” that leads to asthma and rhinitis [7]. The chronicization of AD progresses through damage to the skin barrier due to scratching caused by itchiness [8]. An important feature of AD is the differentiation of biased T helper (Th) 2 cells [9]; the increase in levels of Th2-mediated cytokines and chemokines causes a Th1/Th2 immune imbalance and impairs skin barrier function [6,10,11,12].

Th2 cell activation increases the expression and release of cytokines such as interleukin (IL)-4, IL-5, IL-13, and IL-31 [9]. Additionally, IL-4 has an autocrine function that continuously activates Th2 cells and inhibits Th1 cell activation [13]. Thus, Th1-mediated cytokines, including IL-12 and interferon-γ (IFN-γ), are downregulated resulting in Th1/Th2 immune imbalance. The normal immune system maintains the Th1/Th2 balance with antagonism between IL-4 and IFN-γ [14]. IL-4 and IL-13 promote isotype conversion in B cells that increases the production of immunoglobulins (Ig) E and G1 [6,15]. IgE is detected for AD diagnosis as it is observed at a high concentration in the serum of patients with AD [16,17]. Histamine, a biological response modulator, is released from degranulated mast cells and functions together with IgE to cause itching [5,18]. IL-31 impairs the skin barrier function by causing itching and inhibiting apoptosis in eosinophils [19,20,21].

AD skin lesions express and secrete cytokines IL-25, IL-33, and thymic stromal lymphopoietin (TSLP) [22]. IL-25, IL-33, and TSLP can stimulate Th2 cells directly or indirectly by stimulating dendritic cells (DCs), mast cells, and eosinophils [23]. TSLP activates DCs to induce naive T cell proliferation and primes Th2 cells for differentiation, which is followed by the secretion of high levels of IL-4, IL-5, IL-13, and tumor necrosis factor-α (TNF-α) [24]. Activation of mast cells and eosinophils releases AD-related inflammatory factors such as IL-4, IL-5, IL-6, IL-31, and monocyte chemoattractant protein-1 (MCP-1) [25,26,27]. MCP-1 is reported to be involved in the activation of Th2 cells in mice [28]. Chemokines such as thymus and activation-regulating chemokines (TARC) and macrophage-derived chemokines (MDC) are expressed and secreted in keratinocytes that bind to the C-C motif chemokine receptor 4 (CCR4) of Th2 cells to induce Th2 cell activation [29]. Moreover, regulated on activation, normal T cell expressed and secreted (RANTES) is involved in the degranulation and infiltration of eosinophils [30]. The Th2-mediated cytokine IL-5 induces chronicization of AD through the continuous survival and differentiation of eosinophils [9,31].

The skin barrier blocks the influx of external antigens and allergens; therefore, the maintenance of a healthy skin barrier function is vital for moisture homeostasis and environmental protection [32,33]. Specifically, filaggrin (FLG), involucrin (IVL), and loricrin (LOR) are the key skin barrier proteins components [34]. FLG has a vital role in skin barrier maintenance and constitutes the outermost barrier through the aggregation of the keratinocyte matrix [35]. FLG exists as pro-filaggrin (pro-FLG) and is converted to FLG through dephosphorylation and proteolysis by serine proteases [36]. FLG then participates in epidermal differentiation; following its decomposition into free amino acids, FLG is decomposed into components of natural moisturizing factors (NMF), such as sodium pyrrolidone carboxylic acid and urocanic acid [37]. IVL and LOR constitute, in part, the outer wall of keratinocytes and promote the final differentiation of the epidermis [35]. IVL forms a scaffold in which other proteins are cross-linked [38], while LOR is an insoluble protein in the cornified cell envelope and accounts for 80% of the total protein [39]. LOR binds to IVL and acts as the major protein in the outer skin barrier [40,41]. Tight junction (TJ) proteins are located under the epidermis and contribute to the selective permeability of various substances, including cytokines and hormones. TJ proteins bind to plasma proteins inside the cell, thus forming a loop structure outside the cell that connects with adjacent cells [32]. TJs are formed and interact with proteins such as occludin (OCC) and zonula occludens-1 (ZO-1) [42]. OCC promotes cell adhesion of neighboring cells [43], while ZO-1 binds to several TJ components and interacts with signaling proteins such as heterotrimeric G-proteins [44].

Defects in the skin barrier function are associated with Th2-mediated cytokines, including IL-4, IL-13, and IL-31 [45]. Meanwhile, recent studies have also shown that Th17 cells and Th22-mediated cytokines IL-17 and IL-22 impair the skin barrier function [46]. Th2-mediated cytokines can further activate Th17 and Th22 cells to promote defects in skin barrier function [47].

AD is traditionally treated with topical corticosteroids (TCS); however, long-term use causes topical and systemic side effects [48]. Topical calcineurin inhibitors have been approved to treat AD as replacements for corticosteroids [49], though side effects such as skin malignancies, lymphoma, and leukemia have been reported [50]. Therefore, research on AD therapeutic agents with fewer side effects is required to improve patient treatment and quality of life.

In NC/Nga mice, skin lesions similar to AD appear naturally in a typical environment and are widely used to study mechanisms related to AD [3,51]. 2,4-Dinitrochlorobenzene (DNCB) quickly penetrates the epidermis and causes an increase in IgE, which can lead to hypersensitivity of the skin [52]. Repetitive DNCB application under specific pathogen-free (SPF) conditions causes AD-like skin lesions in the skin of NC/Nga mice [35,51].

*Lithospermum erythrorhizon* (LE) has traditionally been used as a natural preparation in various East Asian countries, including Korea, China, and Japan [53]. Active ingredients of LE include naphthoquinones β, β-dimethyl acryl shikonin, lithospermic acid, and acetyl shikonin, which reportedly have antioxidant and anti-inflammatory effects [54]. However, studies on the improvement of AD-related clinical symptoms, Th1/Th2 immune balance, and skin barrier function recovery using LE have not been conducted. Therefore, this study focused on whether LE restores AD-related clinical symptoms, Th1/Th2 immune balance, and skin barrier function in a DNCB-induced NC/Nga mouse model.

## 2. Materials and Methods

### 2.1. Preparation of the LE Extract

Dried roots of LE were collected from Jindo, Jeollanam-do Province, Korea in October, 2017. A voucher specimen (17-10-009) was deposited in the herbarium of Holistic Bio Co., Ltd. (Seongnam, Gyeonggi-do, Korea). LE was prepared as a dried powder from the diluted ethanol extract, which was provided by Nutrex Co., Ltd. (Seongnam, Gyeonggi-do, Korea). For the preparation of the LR extract, 100 g dried LR was extracted with 300 mL 70% ethanol at 85–90 °C for 2 h with stirring. The residual extract was extracted again under the same conditions. These two extracts were combined and lyophilized to yield 33 g LR extract, which contained 0.23% lithospermic acid.

### 2.2. High-Performance Liquid Chromatography (HPLC) Analysis of LE

Approximately 5.0 g of LE extract was weighed, transferred to a 50 mL measuring flask, dissolved in 50% methanol (1:1 deionized water-MeOH), and passed through a 0.45 μm membrane filter to prepare the test solution. Chromatographic separation was performed on an Agilent 1100 separation module equipped with a C18 column (4.6 mm × 250 mm; 5 μm, CAPCELL PAK, Shiseido, Tokyo, Japan). The pump was connected to two mobile phases: A, 0.1% formic acid, and B, acetonitrile in H_2_O (*v/v*), and the elution flow rate was 1.0 mL/min. The mobile phase was consecutively programmed in linear gradients as follows: 0–20 min, 95% A, 5% B; 20–25 min, 80% A, 20% B; 25–50 min, 70% A, 30% B; 50–75 min 0% A, 100% B; 70–75 min 95% A, 5% B. The UV detector was monitored at a wavelength of 320 nm. The injection volume was 10 µL for each sample solution. The column temperature was maintained at 30 °C.

### 2.3. Animals

Four-week-old male NC/Nga mice were provided by Shizuoka Laboratory Center Inc. (Shizuoka, Japan). Mice were housed at 23 ± 3 °C and 55% ± 5% humidity in individually ventilated cages under SPF conditions with 12 h light and dark cycle. The mice were provided with feed (Catalog number 5L79, Central Laboratory Animal, Seoul, Korea) and water ad libitum. Animals were anesthetized with isoflurane, blood was drawn from the inferior vena cava, and then they were euthanized using CO_2_. Blood was stored at 25 ± 5 °C for 1 h and subsequently centrifuged at 3000× *g*, 4 °C for 15 min to collect serum samples. Serum samples were stored at −80 °C until use. All experimental procedures were performed according to the protocol approved by the Kyung Hee University Animal Care and Use Committee guidelines (approval number KHSASP-20-252).

### 2.4. Induction of AD-like Skin Lesions and LE Treatment

AD-like skin lesions were induced by topical application of DNCB (Sigma-Aldrich, St. Louis, MO, USA) to NC/Nga mice as previously described [4,35]. Briefly, after acclimatization for 1 week, the hair on the dorsal side of the NC/Nga mice was removed using an electric shaver. Mice were randomly divided into normal (naïve control), DNCB (negative control), prednisolone (PD; positive control; Sigma-Aldrich, St. Louis, MO, USA), and 50, 100, and 200 mg/kg LE groups, with six mice assigned to each group. To induce AD-like skin lesions, 1% DNCB was dissolved in a mixture of acetone and ethanol (2:3 *v/v*) and applied twice every other day to the shaved dorsal flank (200 µL) and right ear (100 µL). After sensitization, 0.4% DNCB dissolved in a mixture of acetone and olive oil (3:1 *v/v*) was applied to the dorsal skin (150 μL) and right ear (50 μL) three times a week for 14 weeks. After 9 weeks of induction, mice in the normal and DNCB groups were orally administered 0.5% carboxymethyl cellulose (0.5% CMC) for 4 weeks. PD (3 mg/kg prednisolone) and LE (50, 100, and 200 mg/kg) were orally administered daily for 4 weeks. CMC was used to dissolve PD and LE and was administered to normal and DNCB groups.

### 2.5. Dermatitis Score and Ear Thickness

The dermatitis score was recorded three times per week as previously described [55]. The score grades were 0 (none), 1 (mild), 2 (moderate), or 3 (severe) and were measured for each of the five symptoms tested (e.g., erythema, dryness, maceration, abrasion, and lichenification). The total dermatitis score was quantified as the sum of all individual scores for the five symptoms. The ear thickness was gauged on the right ear of each mouse three times a week using a thickness gauge (Mitutoyo Corporation, Tokyo, Japan).

### 2.6. Scratching Behavior

Scratching behavior was recorded three times per week [56]. After vehicle administration, the mice were acclimated to an acrylic cage for 1 h and the scratching movements around the neck, ears, and dorsal flank skin with hind paws were measured and recorded for 30 min. The scores ranged from 0 to 4 (0 points (none), 2 points (less than 1.5 s), and 4 points (more than 1.5 s)). The total score for scratching behavior was presented as the sum of the individual measurements.

### 2.7. Histological Analysis

The dorsal skin was cut and fixed in 10% formalin, then sliced to a thickness of 4 μm. Tissue sections were stained with hematoxylin and eosin (H&E) and toluidine blue (TB). After staining, images were taken with an optical microscope (400×, DP Controller Software; Olympus Optical, Tokyo, Japan). The epidermis thickness and the number of infiltrated inflammatory cells (e.g., mast cells and eosinophils) were measured at six sites per mouse using Image J software (National Institute of Health, Starkville, MD, USA).

### 2.8. Serum Immunoglobulin and Histamine Assay

The levels of IgE, histamine, IgG1, and IgG2a in the serum were measured using a mouse enzyme-linked immunosorbent assay (ELISA) kit according to the manufacturer’s instructions (IgE, Shibayagi, Gunma, Japan; IgG1 and IgG2a, Enzo Life Sciences, Farmingdale, NY, USA; histamine, Elabscience, Huston, ID, USA).

### 2.9. Serum Cytokines and Chemokines Assay

The serum cytokine and chemokine levels were measured using a mouse ELISA kit, according to the manufacturer’s instructions (IL-4 and IL-6, Enzo Life Sciences, Farmingdale, NY, USA; TSLP, MCP-1, Elabscience, Huston, ID, USA; IL-1β, TARC, MDC, and RANTES, R&D Systems Inc., Minneapolis, MN, USA).

### 2.10. Isolation of Splenocytes and Analysis of Cytokines and Chemokines

Splenocytes were isolated in a sterile environment. Splenocytes were crushed with a sterile syringe plunger and collected using a cell strainer (BD Biosciences, Franklin Lakes, NJ, USA). Subsequently, after treatment with red blood cell lysis buffer, splenocytes were washed three times with RPMI-1640 (Gibco, Carlsbad, NY, USA) supplemented with 10% FBS. The isolated splenocytes were treated with 5 μg/mL concanavalin A (Con-A) (Sigma-Aldrich, St. Louis, MO, USA) and incubated in 24-well plates for 72 h at a concentration of 1 × 10^6^ cells/well at 37 °C, 5% CO_2_. After incubation, supernatants were collected and splenocytes were homogenized in lysis buffer containing cOmplete™ Protease Inhibitor Cocktail tablets (Roche Diagnostics, Indianapolis, IN, USA). The lysate was centrifuged at 10,000× *g* for 10 min at 4 °C. The collected splenocyte supernatants and lysates were frozen at −80 °C for subsequent cytokine analysis. The levels of IL-5, IL-12, IL-13, IL-17, IL-22, IL-25, IL-31, IL-33, TNF-α, and IFN-γ cytokines in the splenocyte supernatant were measured using an ELISA kit according to the manufacturer’s instructions (Elabscience, Houston, ID, USA). The lysate protein concentration was measured using a Pierce™ BCA Protein Assay Kit (Thermo Fisher Scientific, Rockford, IL, USA). The levels of cytokines and chemokines in the supernatants were normalized to the protein concentration of the lysate.

### 2.11. RNA Extraction and Quantitative Real-Time Polymerase Chain Reaction (RT-qPCR)

Total RNA was extracted from mouse dorsal skin samples using the Easy-Red Total RNA Extraction Kit. Chloroform was added, and the mixture was stored at room temperature for 20 min. The supernatant was collected by centrifugation at 10,000× *g* for 15 min at 4 °C. The cells were treated with an equal volume of isopropanol as the supernatant and incubated overnight for 24 h. The mixture was then centrifuged for 15 min at 10,000× *g* and 4 °C and washed with 75% ethanol. After RNA was dissolved in DEPC water, cDNA was synthesized using a cDNA synthesis kit(Takara Korea Biomedical, Inc., Shiga, Japan). Quantitative real-time polymerase chain reaction (RT-qPCR) was performed on an ABI StepOnePlus™ real-time PCR system (Applied Biosystems, Waltham, MA, USA) using the synthesized cDNA as a template and SYBR Premix EX Taq (TaKaRa Bio, Shiga, Japan). The primer sequences are listed in Table 1. Gene expression levels were normalized to GAPDH using the 2^−ΔΔCt^ method for the cycle threshold (Ct) value.

### 2.12. Western Blotting

Dorsal skin tissues were frozen in liquid nitrogen, crushed using a pestle, and subsequently homogenized with lysis buffer containing cOmplete™ Protease Inhibitor Cocktail tablets (Roche Diagnostics, Indianapolis, IN, USA). The lysates were sonicated and centrifuged at 10,000× *g* for 15 min at 4 °C. The protein concentration in the supernatant was quantified using the Pierce™ BCA protein assay kit (Thermo Fisher Scientific, Rockford, IL, USA). After quantitation, equal amounts of protein were loaded into a 12% sodium dodecyl sulfate-polyacrylamide gel (SDS-PAGE; Bio-Rad, CA, USA) for electrophoresis and then transferred to a polyvinylidene fluoride (PVDF) membrane. The membrane was blocked with 5% skim milk in Tris-buffered saline with 0.5% Tween-20 (TBST) and incubated with 1:1000 primary antibody overnight at 4 °C. The following day, the membranes were treated with a horseradish peroxidase-conjugated (HRP) secondary antibody at a dilution of 1:5000 (GeneTex, Inc., Irvine, CA, USA) for 2 h and were visualized using a ChemiDoc™XRS + System (Bio-Rad, Richmond, CA, USA). The expression level of each protein was analyzed using Image Lab statistical software (Bio-Rad, Richmond, CA, USA) and normalized to β-actin. The primary antibodies used for Western blotting were as follows: filaggrin (FLG), occluding (OCC), and loricrin (LOR) (GeneTex, Inc., Irvine, CA, USA), involucrin (IVL), β-actin (Santa Cruz, CA, USA), and zonula occludens-1 (ZO-1) (Abcam, Cambridge, MA, USA).

### 2.13. Statistical Analysis

Data are presented as means ± standard deviation (SD). Statistical analysis was performed using one-way analysis of variance (ANOVA) and Tukey’s honestly significant difference test. Statistically significant differences were evaluated using SPSS (SPSS Inc., Chicago, IL, USA).

## 3. Results

### 3.1. Identification and Qualification of Lithospermic Acid in the LE Extracts

HPLC analysis at 320 nm was used to determine the components of LE, and the established standard chromatogram is shown in Figure 1. Lithospermic acid is highlighted, which is a major constituent of LE, and was detected in high concentrations in our LE samples. The retention time of lithospermic acid was 41.118 ± 0.21 min. The content analysis indicated that LE contained 1.34 ± 0.009 mg/g of lithospermic acid.

### 3.2. LE Attenuates DNCB-Induced AD-like Symptoms and Scratching Behavior

The schedule for DNCB induction and LE oral administration is presented in Figure 2A. Figure 2B shows representative images of the six test groups, including the normal, DNCB, PD, and 50, 100, and 200 mg/kg LE groups, on the last day of the 14th week of the experiment. Repetitive application of DNCB aggravated the clinical symptoms associated with AD, including erythema, maceration, lichenification, abrasion, dryness, and scratching behavior, over the first nine weeks. However, administration of LE and PD reduced these lesions, and the dermatitis score, ear thickness, and scratching behavior were significantly higher in the DNCB group compared to the normal group; however, LE administration significantly reduced these parameters in a dose-dependent manner (Figure 2C). The dermatitis score and scratching behavior showed a similar efficacy between the 50 mg/kg LE group and the positive control PD group, and ear thickness measurements showed a similar effect between the 100 mg/kg LE group and the PD group. Taken together, LE attenuated the dermatitis score, ear thickness, and scratching behavior in the AD mouse model.

### 3.3. LE Reduces Epidermal Thickening as Well as Eosinophil and Mast Cell Infiltration

To evaluate how inflammatory responses contribute to AD symptoms, we measured epidermis thickness and amount of eosinophil and mast cell infiltration, both of which were significantly higher in the DNCB group compared to the normal group. However, LE treatment reduced the epidermal thickness and the number of infiltrated inflammatory cells in a dose-dependent manner, showing a similar efficacy between the 100 mg/kg LE group and the PD group (Figure 3).

### 3.4. LE Decreases IL-4, IgE, and Histamine Serum Levels

IL-4, IgE, and histamine serum levels were investigated to evaluate the effect of LE on AD itching. We found that the IL-4, IgE, and histamine levels were significantly higher in the DNCB group compared to the normal group. However, LE treatment reduced these levels in a dose-dependent manner, with similar efficacy in reducing IL-4 levels observed between the 100 mg/kg LE and PD groups. However, the LE group showed a higher efficacy in reducing IgE and histamine levels than the PD group (Figure 4A–C). These results show that LE reduced cytokines, IgE, and histamine in serum related to itching.

### 3.5. LE Restores Th1/Th2 Immune Balance by Regulating Serum IgG1 and IgG2a Levels in NC/Nga Mice

The restoration of Th1/Th2 balance was investigated by measuring the levels of IgG1 and IgG2a in response to LE treatment. Serum IgG1 levels were higher in the DNCB group compared to the normal group; however, LE treatment reduced the IgG1 levels in a dose-dependent manner, with a similar efficacy observed between the 100 mg/kg LE and PD groups. Moreover, the IgG2a levels were higher in the DNCB group than in the normal group. Meanwhile, compared to the DNCB group, the LE group increased IgG2a levels in a dose-dependent manner, whereas IgG2a levels decreased in the PD group. These results showed LE treatment restored the IgG1/IgG2a ratio in a dose-dependent manner, with a similar efficacy between the 100 mg/kg LE and PD groups (Figure 5A–C).

### 3.6. LE Decreases the Levels of AD-Related Cytokines and Chemokines in NC/Nga Mouse Serum

To evaluate the effects of LE on immune responses in the AD mouse model, we measured the levels of IL-1β, IL-6, TNF-α, TSLP, TARC, MDC, RANTES, and MCP-1. The levels of cytokines and chemokines in the serum of NC/Nga mice were significantly higher in the DNCB group than in the normal group. However, LE treatment decreased the levels of AD-related cytokines and chemokines in the serum in a dose-dependent manner and showed a similar efficacy between the 100 mg/kg LE and PD groups (Figure 6A–C). These results suggest that LE alleviated AD by reducing the levels of AD-related cytokines and chemokines in the serum.

### 3.7. LE Regulates the Balance of Cytokines and Chemokines Secretion in Splenocytes

We investigated the levels of cytokines and chemokines secreted from splenocytes to confirm the anti-atopic effect of LE. The levels of the Th2-mediated cytokines IL-4, IL-5, IL-13, and IL-31, and the cytokines IL-25, IL-33, and TSLP, which indirectly activate Th2, significantly increased in the DNCB group compared to the normal group. In addition, AD-related cytokines and chemokines, namely, IL-1β, IL-6, IL-17, IL-22, TNF-α, and MCP-1, were significantly higher in the DNCB group compared to the normal group. However, LE treatment reduced the AD-related cytokine and chemokine levels in a dose-dependent manner and showed a similar efficacy between the 100 mg/kg LE and PD groups (Figure 7A–E). Additionally, the levels of Th1-mediated cytokines IL-12 and IFN-γ were significantly reduced in the DNCB group compared to the normal group. Meanwhile, LE treatment restored these cytokine levels in a dose-dependent manner. Interestingly, the levels of Th1-mediated cytokines decreased in the PD group (Figure 7F).

### 3.8. LE Inhibits the Gene Expression of Cytokines, Chemokines, and CCR4 Involved in Th2 Activation in the Dorsal Skin

The gene expression levels of AD-related cytokines and chemokines that directly activate Th2 cells were investigated in the dorsal skin of NC/Nga mice. The expression of IL-25, IL-33, TSLP, RANTES, TARC, MDC, and CCR4 was significantly higher in the DNCB group than in the normal group. However, LE treatment reduced gene expression in a dose-dependent manner and showed a similar efficacy between the 100 mg/kg LE and PD groups (Figure 8).

### 3.9. LE Restores the Balance of AD-Related Cytokine and Chemokine Gene Expression in the Dorsal Skin of the NC/Nga Mouse

We investigated the levels of cytokines and chemokines expressed in the dorsal skin to confirm the anti-atopic effect of LE. Gene expression of the Th2-mediated cytokines IL-4, IL-5, IL-13, and IL-31 was significantly higher in the DNCB group compared to the normal group. In addition, gene expression of the AD-related cytokines IL-1β, IL-6, TNF-α, IL-17, and IL-22 was significantly increased in the DNCB group compared to the normal group. LE treatment reduced the expression of AD-related cytokines and chemokines, including Th2-mediated cytokines, in a dose-dependent manner, and showed a similar efficacy between the 100 mg/kg LE and PD groups (Figure 9A–C).

Expression of Th1-mediated cytokine (IL-12 and IFN-γ) genes was significantly reduced in the DNCB group compared to the normal group, while LE restored their expression in a dose-dependent manner. We also found that the PD group showed reduced expression of Th1-mediated cytokines compared to that in the DNCB group (Figure 9D).

### 3.10. LE Restores Defects in Skin Barrier Function Caused by AD

In the dorsal skin of NC/Nga mice, pro-FLG gene expression and the abundance of proteins responsible for skin barrier function were investigated. The expression of pro-FLG was significantly reduced in the DNCB group compared to the normal group; however, LE treatment restored its expression in a dose-dependent manner and showed a similar efficacy between the 100 mg/kg LE and PD treatment groups (Figure 10B). The abundance skin barrier proteins FLG, IVL, and LOR, and the TJ proteins OCC and ZO-1, was significantly reduced in the DNCB group compared to the normal group, while LE treatment restored the expression of skin barrier and TJ-related proteins in a dose-dependent manner. The skin barrier proteins FLG, IVL, and LOR were present at similar levels when samples were treated with 100 mg/kg LE and PD (Figure 10A,C–E). However, the expression of the TJ proteins OCC and ZO-1 was not recovered in the PD group compared to the DNCB group (Figure 10A,F,G).

## 4. Discussion

AD induces biased differentiation of Th2 cells, resulting in Th1/Th2 immune imbalance, followed by increased itchiness and complex pathophysiological changes that lead to the skin barrier function damage [10,35]. In the present study, the efficacy of LE extract against AD was investigated using an in vivo animal model, focusing on whether LE could improve AD-related clinical symptoms, Th1/Th2 immune balance, and restoration of skin barrier function. IgE secretion is increased in B cells by the Th2-mediated cytokine IL-4 [57], and the degranulation of mast cells by IgE increases itching through the release of histamine [27,58]. Continuous scratching can increase erythema, abrasions, and erosions, along with lichenification, which thickens the skin [59,60,61]. In addition, damage to the skin barrier caused by this scratching can further cause loss of moisture, leading to dry skin [62]. LE treatment led to improved AD-related clinical symptoms and reduced infiltration of inflammatory cells in a dose-dependent manner (Figure 2B–E).

The serum secretion levels of IL-4, IgE, and histamine decreased in a dose-dependent manner following LE treatment compared to the DNCB group (Figure 4). Therefore, LE appears to reduce infiltration and degranulation of mast cells, thereby reducing histamine release, as well as scratching, ultimately leading to improved clinical symptoms, including erythema, erosion, and abrasion. In addition, IL-4 reduced the expression of skin barrier proteins [45], further indicating that the dry skin symptoms were improved through the restoration of the skin barrier protein. LE showed a greater inhibitory effect on IL-4, IgE, and histamine than PD, which is used as a clinical treatment for AD (Figure 4).

Eosinophils increase in both the blood and skin of AD patients and are reportedly involved in chronic AD while playing an important role in the Th2 immune response [63]. LE treatment reduced the number of infiltrating eosinophils in a dose-dependent manner (Figure 3A(A-a,A-b)) and lichenification was recovered by LE, thus demonstrating its ability to improve a clinical symptom of chronic AD. The recovery of lichenification was consistent with a decrease in the number of infiltrated eosinophils and epidermal thickness.

In the keratinocytes of AD patients, the expression and secretion of cytokines IL-25, IL-33, and TSLP, and the chemokines TARC, MDC, and RANTES, which are directly or indirectly related to the activation of Th2 cells, are increased [64]. In addition, expression of CCR4, a receptor for TARC and MDC, increases on the surface of Th2 cells, thereby increasing Th2 cell invasion and activity [29,65]. LE dose-dependently reduced the levels of TSLP, TARC, MDC, and RANTES in the serum and gene expression levels of AD-related IL-25, IL-33, TSLP, TARC, MDC, and RANTES, including CCR4, in the dorsal skin of NC/Nga mice (Figure 6E–H and Figure 8). In addition, LE dose-dependently decreased the levels of cytokines and chemokines, such as IL-25, IL-33, and TSLP, secreted by splenocytes in the AD mouse model (Figure 7B). These results suggest that LE reduces the expression and secretion of cytokines and chemokines that directly or indirectly activate Th2 cells in the serum, splenocytes, and dorsal skin. Inhibition of CCR4 expression reduced the binding of TARC and MDC, suggesting that the activity of Th2 cells was suppressed by LE treatment. The decreased expression and secretion of RANTES in the dorsal skin indicated that LE inhibited the progression to chronic AD. These results are consistent with the reduction in infiltrated eosinophils and restoration of lichenification (Figure 2B and Figure 3A(A-b)).

IL-25, IL-33, and TSLP are involved in the activation of several immune cells, including mast cells and eosinophils, and induce the secretion of Th2-mediated cytokines [66]. Activation of mast cells and eosinophils is involved in allergic responses, in which gene expression and secretion of AD-related cytokines and chemokines such as IL-4, IL-5, IL-6, IL-13, IL-31, and MCP-1 increase [27,67,68]. We found that LE had a dose-dependent effect on the serum levels of MCP-1 as well as MCP-1 and Th2-mediated cytokines IL-4, IL-5, IL-13, and IL-31 secreted by splenocytes (Figure 6H and Figure 7A,C). Moreover, LE treatment downregulated the expression of IL-4, IL-5, IL-13, and IL-31 in dorsal skin in a dose-dependent manner (Figure 9A). By suppressing the activity of immune cells, LE could resolve the immune imbalance in the AD mouse model via suppression of Th2 mediated cytokines IL-4 and IL-13, which continuously activate Th2 cells in an autocrine manner [69].

IL-5 and IL-31 have been reported as cytokines that delay the increase and apoptosis of eosinophils, while MCP-1, which is primarily expressed and secreted by eosinophils, is involved in the invasion of Th2 cells [70,71]. LE reduced both the gene expression and secreted levels of IL-5, IL-31, and MCP-1 (Figure 6H, Figure 7A,C and Figure 9A), indicating that LE could reduce the number of infiltrated eosinophils, consistent with improved clinical symptoms and lichenification.

IL-12 and IFN-γ are required for the differentiation of Th1 cells [72]. IL-12 promotes the differentiation of Th1 cells and IFN-γ secreted from Th1 cells suppresses the secretion of IL-4, a Th2-mediated cytokine, to suppress the production of IgE and IgG1 and promote the production of IgG2a in B cells [4,73]. Therefore, the IgG1/IgG2a ratio is used as a representative marker of Th1/Th2 balance [4]. In the splenocytes and dorsal skin of NC/Ng mice, secretion and gene expression of IL-12 and IFN-γ were decreased in the DNCB group compared to the normal group and recovered in a dose-dependent manner by LE treatment (Figure 7F and Figure 9D). In the LE group, IgG1 decreased and IgG2a was restored, resulting in a dose-dependent recovery of the Th1/Th2 ratio (Figure 5). In contrast, PD was shown to restore the Th1/Th2 ratio by decreasing the levels of both Th1 and Th2 cells associated immunoglobulins. Taken together, our results indicate that PD can restore the Th1/Th2 immune balance by inhibiting the activity of both Th1 and Th2 cells, whereas LE restored Th1/Th2 immune balance by regulating the activation of Th1 and Th2 cells.

IL-1β, IL-6, and TNF-α are expressed and secreted by many inflammatory cells, including keratinocytes, Th2 cells, and mast cells; they are involved in the progression of acute and chronic AD and activate Th17 and Th22 cells [74,75]. LE treatment decreased the levels of IL-1β, IL-6, and TNF-α in the serum and splenocytes (Figure 6A–C and Figure 7D), and the gene expression levels of IL-1β, IL-6, and TNF-α in the dorsal skin in a dose-dependent manner (Figure 9B). These results show that LE prevents the progression of acute and chronic AD through the regulation of both gene expression and protein abundance of cytokines involved in Th cell activation. These findings also suggest that LE may be involved in the restoration of immune balance by regulating the activity of Th17 and Th22 cells.

Th17 and Th22 cells show a positive correlation with Th2-mediated cytokines [47,76]. Th17 cells produce IL-17 and IL-22, while Th22 cells increase the expression and secretion of IL-22 [4,45]. IL-17 and IL-22 increase the differentiation of Th2 cells and impair skin barrier function [45]. LE decreased the expression and secretion levels of IL-17 and IL-22 in the dorsal skin and splenocytes (Figure 7E and Figure 9C), which reduced the activity of Th17 and Th22 cells. This suggests that LE is involved in the restoration of immune balance and skin barrier function through the deactivation of Th17 and Th22 cells.

Cytokines and chemokines associated with AD, such as IL-4, IL-6, IL-13, IL-17, IL-31, IL-22, IL-33, TSLP, and MCP-1, downregulate the abundance of pro-FLG and as well as that of the skin barrier-related proteins FLG, IVL, and LOR, and the TJ proteins OCC and ZO-1 [4,31,35,47,77,78], resulting in impaired skin barrier function. It has been reported that reduction of FLG, IVL, and LOR aggravates the Th2 immune response by inducing an increase in microbiome species, such as *Staphylococcus aureus* in the skin [79]. In this study, the gene expression of pro-FLG and the abundance of skin barrier-related proteins FLG, IVL, and LOR decreased in the DNCB group compared to the normal group. However, pro-FLG gene expression and FLG, IVL, and LOR protein abundance were recovered in a dose-dependent manner by LE treatment, and slightly recovered in the PD group (Figure 10B–E). Some studies have reported that TCS increases skin barrier proteins such as FLG and LOR [80]. The restoration of pro-FLG, FLG, IVL, and LOR by LE treatment suggests that the clinical symptoms and immune balance associated with AD were recovered.

The TJ region is located below the stratum corneum, sealing the space between cells and binding cells to each other [62]. It has been reported that AD-induced TJ damage induces expansion of Langerhans cell dendrites into the TJ, triggering a Th2-mediated immune response [81]. As TJs exhibit an inverse correlation with the progression of AD, we hypothesized that TJ damage is an important feature of AD [4]. The expression of the TJ proteins OCC and ZO-1 were lower in the DNCB group compared to the normal group, however it was restored in a dose-dependent manner with LE treatment. Meanwhile, the abundance of TJ proteins was not increased in the PD group compared to the DNCB group (Figure 10F,G). A previous study reported that TCS negatively affects cell permeability by decreasing the abundance of TJ proteins, including OCC [82]. Unlike TCS, LE treatment might alleviate AD by restoring skin barrier function. Taken together, these results suggest that LE is a potential anti-AD drug candidate that functions by restoring the Th1/Th2 immune balance and enhancing skin barrier function.

## 5. Conclusions

Oral administration of LE ameliorated AD-related clinical symptoms, including dermatitis score, ear thickness, scratching behavior, epidermal thickness, and infiltration of inflammatory cells, in the NC/Nga AD mouse model. In addition, LE reduces scratching behavior by inhibiting the release of IgE and histamine in NC/Nga mice. LE also reduces the levels of AD-related cytokines and chemokines in the serum and restores the Th1/Th2 immune balance via regulation of IgG1 and IgG2a. The immune balance was restored through regulation of gene expression and secretion of AD-related cytokines and chemokines derived from Th1/Th2/Th17/Th22 in splenocytes and dorsal skin. Furthermore, LE increased the expression of proteins responsible for skin barrier function, restoring TJ proteins to a greater extent than PD treatment. Taken together, the findings of this study suggest that LE is a potential therapeutic agent that can alleviate AD by regulating immune balance and restoring skin barrier function.

## Figures and Tables

**Figure 1 nutrients-13-03209-f001:**
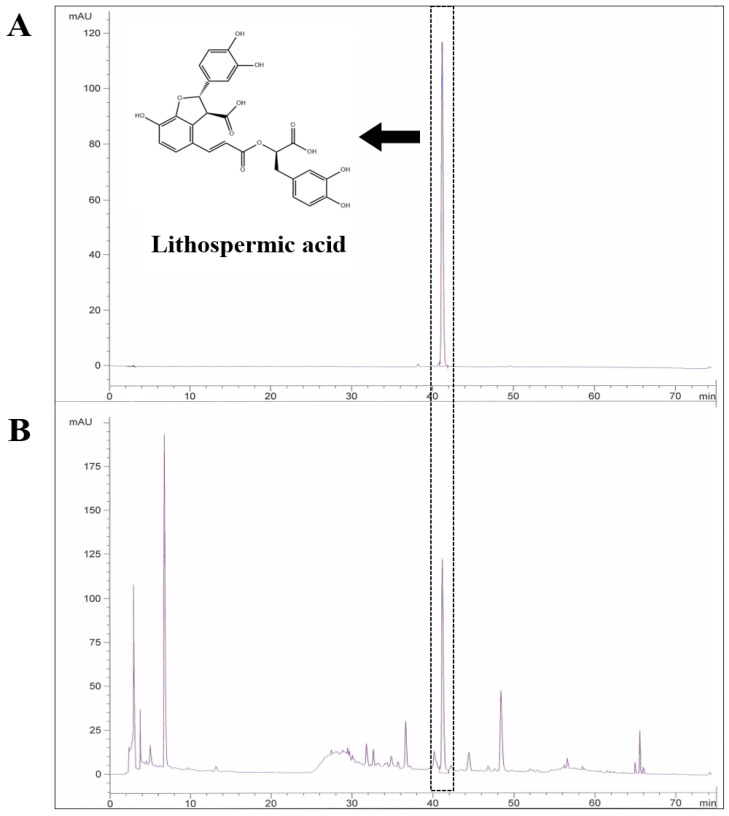
Representative HPLC chromatogram of LE. HPLC chromatograms of (**A**) lithospermic acid and (**B**) the LE extract. The arrows represent lithospermic acid. The retention time of lithospermic acid was 41.118 ± 0.21 min.

**Figure 2 nutrients-13-03209-f002:**
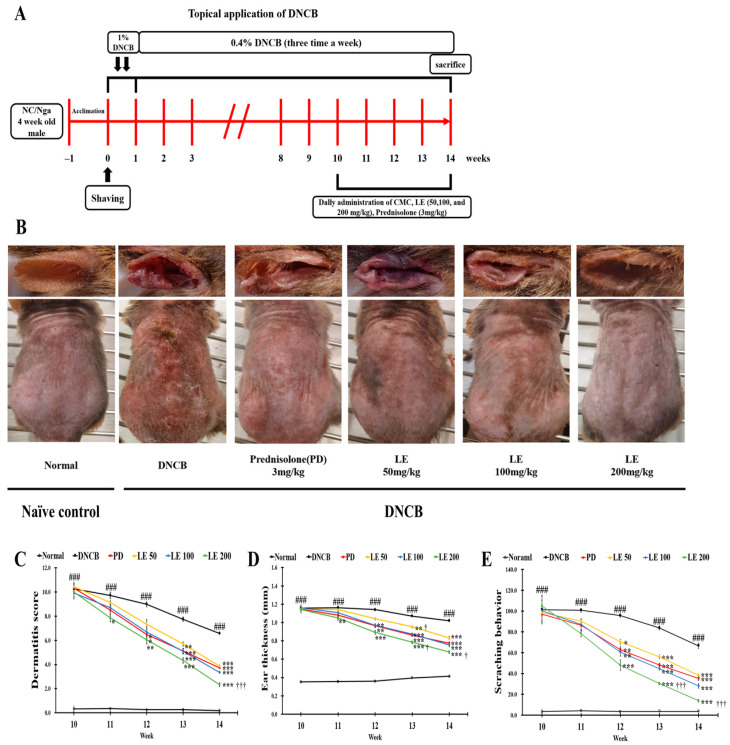
Experimental procedure and effects of LE on the clinical features of AD-like symptoms induced by DNCB in NC/Nga mice. (**A**) Schematic diagram of the experimental procedure for AD lesion induction and LE treatment. (**B**) On the last day of the 14th week of the experiment, the right ear and dorsal skin of mice representing each group were shown. (**C**–**E**) The clinical features of NC/Nga mice (**C**) dermatitis score, (**D**) ear thickness, and (**E**) scratching behavior were evaluated three times a week in the term of the administration of CMC, PD, and LE. The results were expressed as means ± SD (*n* = 6). ^###^ *p* < 0.001 vs. normal (DNCB untreated group), * *p* < 0.05, ** *p* < 0.01, and *** *p* < 0.001 vs. DNCB (negative control; DNCB treated group), PD (positive control; prednisolone 3 mg/kg) treatment group, LE (LE 50, 100 or 200 mg/kg) treatment group, and ^†^ *p* < 0.05, ^†††^ *p* < 0.001 vs. PD (positive control; prednisolone 3 mg/kg) treatment group.

**Figure 3 nutrients-13-03209-f003:**
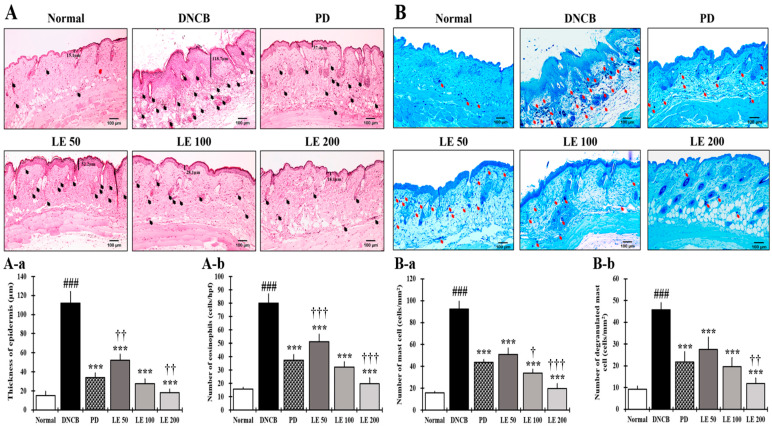
Effect of LE on DNCB-induced histological features of AD-like symptoms in NC/Nga mice. (**A**,**B**) Representative histological images of dorsal skin from mice treated with DNCB (400×, scale bar = 100 μm). (**A**) H&E staining of dorsal skin. The epidermis thickness is indicated by bars and black arrows indicate the infiltration of eosinophils into the skin. (**A-a**) The thickness of the epidermis was measured and averaged from the dorsal skin lesions. (**A-b**) The number of eosinophils infiltrated the dorsal skin. (**B**) Toluidine blue staining of dorsal skin. The red arrows indicate the penetration of mast cells into the skin. (**B-a**) Mast cells present in dorsal skin lesions. (**B-b**) The number of mast cells that infiltrated the dorsal skin lesion. All data were collected from six random dorsal skin sites for each mouse. The results are expressed as means ± SD (*n* = 6). ^###^ *p* < 0.001 vs. normal (DNCB untreated group), *** *p* < 0.001 vs. DNCB (negative control; DNCB treated group), PD (positive control; prednisolone 3 mg/kg) treatment group, LE (LE 50, 100 or 200 mg/kg) treatment group, and ^†^ *p* < 0.05, ^††^ *p* < 0.01, and ^†††^ *p* < 0.001 vs. PD (positive control; prednisolone 3 mg/kg) treatment group.

**Figure 4 nutrients-13-03209-f004:**
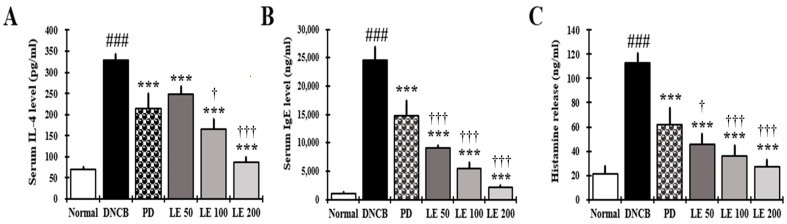
Effect of LE on IL-4, IgE, and histamine levels in the serum of NC/Nga mice. The levels of (**A**) IL-4, (**B**) IgE, and (**C**) histamine in the serum. Serum was collected on the last day of the experiment and measured using ELISA. The results were expressed as means ± SD (*n* = 6). ^###^ *p* < 0.001 vs. normal (DNCB untreated group), *** *p* < 0.001 vs. DNCB (negative control; DNCB treated group), PD (positive control; prednisolone 3 mg/kg) treatment group, LE (LE 50, 100 or 200 mg/kg) treatment group, and ^†^ *p* < 0.05, ^†††^ *p* < 0.001 vs. PD (positive control; prednisolone 3 mg/kg) treatment group.

**Figure 5 nutrients-13-03209-f005:**
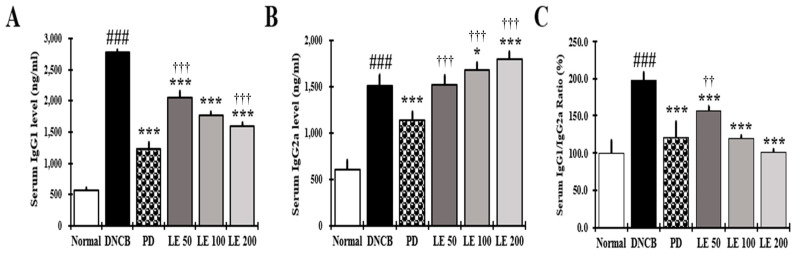
Effect of LE on the levels of IgG1, IgG2a, and the IgG1/IgG2a ratio in the serum of NC/Nga mice. The levels of (**A**) IgG1, (**B**) IgG2a, and (**C**) IgG1/IgG2a ratio (%) in the serum. Serum was collected on the last day of the experiment and measured using ELISA. The results were expressed as means ± SD (*n = 6*). ^###^ *p* < 0.001 vs. normal (DNCB untreated group), * *p* < 0.05, *** *p* < 0.001 vs. DNCB (negative control; DNCB treated group), PD (positive control; prednisolone 3 mg/kg) treatment group, LE (LE 50, 100, or 200 mg/kg) treatment group, and ^††^ *p* < 0.01, ^†††^ *p* < 0.001 vs. PD (positive control; prednisolone 3 mg/kg) treatment group.

**Figure 6 nutrients-13-03209-f006:**
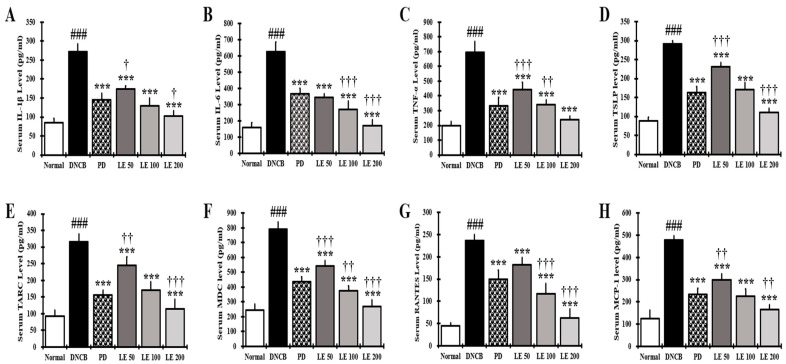
Effect of LE on AD-related cytokines and chemokines in the serum of NC/Nga mice. (**A**–**H**) The levels of (**A**) IL-1β, (**B**) IL-6 (**C**) TNF-α, (**D**) TSLP, (**E**) TARC (**F**) MDC, (**G**) RANTES, and (**H**) MCP-1 in the serum. Serum was collected on the last day of the experiment and measured using ELISA. The results were expressed as means ± SD (*n* = 6). ^###^ *p* < 0.001 vs. normal (DNCB untreated group), *** *p* < 0.001 vs. DNCB (negative control; DNCB treated group), PD (positive control; prednisolone 3 mg/kg) treatment group, LE (LE 50, 100 or 200 mg/kg) treatment group, and ^†^ *p* < 0.05, ^††^ *p* < 0.01, and ^†††^ *p* < 0.001 vs. PD (positive control; prednisolone 3 mg/kg) treatment group.

**Figure 7 nutrients-13-03209-f007:**
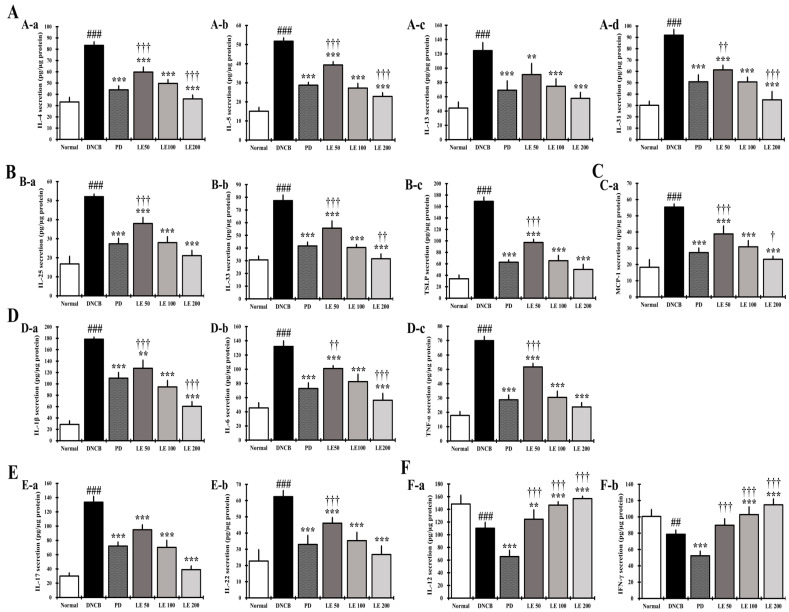
Effect of LE on AD-related cytokine and chemokine secretion in NC/Nga mouse splenocytes. (**A**) The levels of (**A-a**) IL-4, (**A-b**) IL-5, (**A-c**) IL-13, and (**A-d**) IL-31 in the supernatant of splenocytes. (**B**) The levels of (**B-a**) IL-25, (**B-b**) IL-33, and (**B-c**) TSLP in the supernatant of splenocytes. (**C**) The levels of (**C-a**) MCP-1 in the supernatant of splenocytes. (**D**) The levels of (**D-a**) IL-1β, (**D-b**) IL-6, and (**D-c**) TNF-α in the supernatant of splenocytes. (**E**) The levels of (**E-a**) IL-17 and (**E-b**) IL-22 in the supernatant of splenocytes. (**F**) The levels of Th1-mediated cytokines (**F-a**) IL-12 and (**F-b**) IFN-γ. Splenocytes from NC/Nga mice were stimulated with Con-A for 72 h, and then the supernatant was measured using ELISA. Cytokines and chemokines were normalized to the protein concentration of the lysate. The results were expressed as means ± SD (*n* = 6). ^##^ *p* < 0.01, ^###^ *p* < 0.001 vs. normal (DNCB untreated group), ** *p* < 0.01, *** *p* < 0.001 vs. DNCB (negative control; DNCB treated group), PD (positive control; prednisolone 3 mg/kg) treatment group, LE (LE 50, 100 or 200 mg/kg) treatment group, and ^†^ *p* < 0.05, ^††^ *p* < 0.01, and ^†††^ *p* < 0.001 vs. PD (positive control; prednisolone 3 mg/kg) treatment group.

**Figure 8 nutrients-13-03209-f008:**
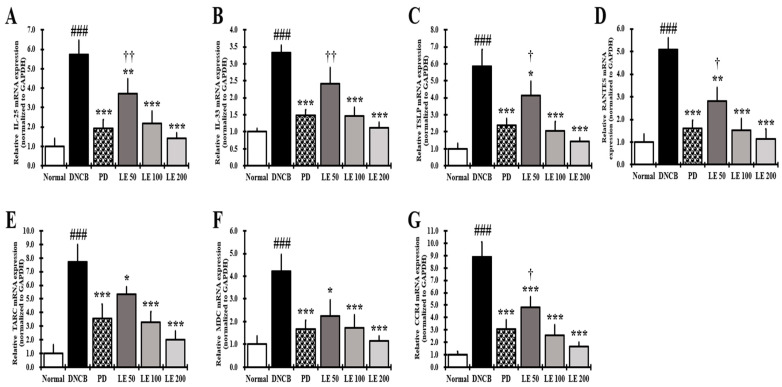
Effect of LE on gene expression of cytokines, chemokines, and CCR4 that activate Th2 cells in dorsal skin of NC/Nga mice. Gene expression of (**A**) IL-25, (**B**) IL-33, (**C**) TSLP, (**D**) RANTES, (**E**) TARC, (**F**) MDC, and (**G**) CCR4. Total RNA was extracted from the dorsal skin of NC/Nga mice and normalized to GAPDH. The results were expressed as means ± SD (*n* = 6). ^###^ *p* < 0.001 vs. normal (DNCB untreated group), * *p* < 0.05, ** *p* < 0.01, and *** *p* < 0.001 vs. DNCB (negative control; DNCB treated group), PD (positive control; prednisolone 3 mg/kg) treatment group, LE (LE 50, 100 or 200 mg/kg) treatment group, and ^†^ *p* < 0.05, ^††^ *p* < 0.01 vs. PD (positive control; prednisolone 3 mg/kg) treatment group.

**Figure 9 nutrients-13-03209-f009:**
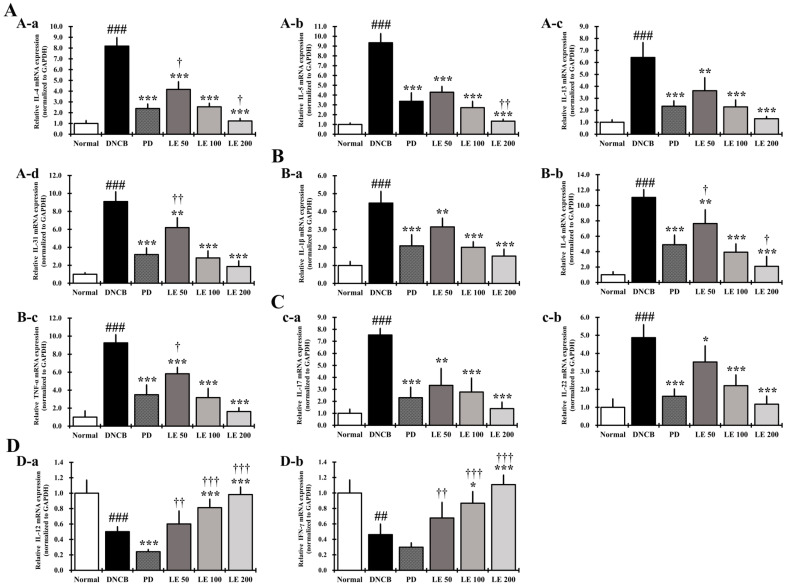
Gene expression of AD-related cytokines and chemokines in response to LE treatment of the dorsal skin of NC/Nga mice. (**A**) Gene expression levels of (**A-a**) IL-4, (**A-b**) IL-5, (**A-c**) IL-13, and (**A-d**) IL-31. (**B**) Gene expression levels of (**B-a**) IL-1β, (**B-b**) IL-6, and (**B-c**) TNF-α. (**C**) Gene expression levels of (**c-a**) IL-17 and (**c-b**) IL-22. (**D**) Gene expression levels of (**D-a**) IL-12 and (**D-b**) IFN-γ. Total RNA was extracted from the dorsal skin of NC/Nga mice and normalized to GAPDH. The results were expressed as means ± SD (*n* = 6). ^##^ *p* < 0.01, ^###^ *p* < 0.001 vs. normal (DNCB untreated group), * *p* < 0.05, ** *p* < 0.01, and *** *p* < 0.001 vs. DNCB (negative control; DNCB treated group), PD (positive control; prednisolone 3 mg/kg) treatment group, LE (LE 50, 100 or 200 mg/kg) treatment group, and ^†^ *p* < 0.05, ^††^ *p* < 0.01, and ^†††^ *p* < 0.001 vs. PD (positive control; prednisolone 3 mg/kg) treatment group.

**Figure 10 nutrients-13-03209-f010:**
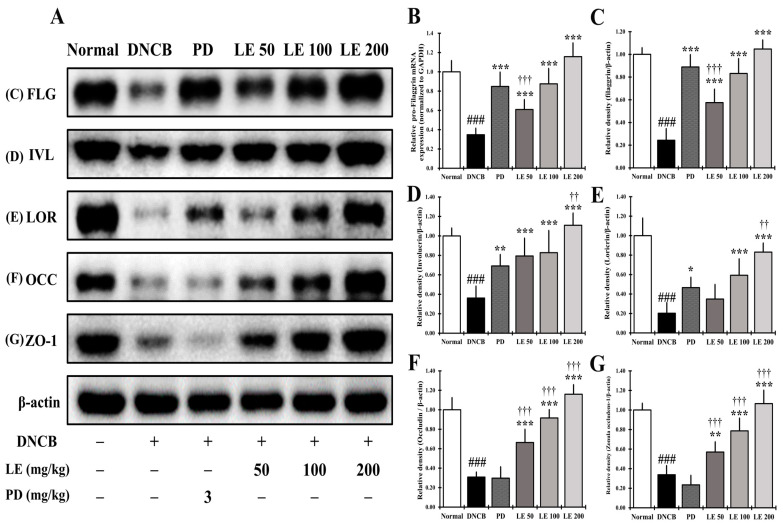
Restoration effect of LE on the skin barrier proteins in the dorsal skin of NC/Nga mice. (**A**) Abundance of proteins related to skin barrier function in the dorsal skin of NC/Nga mice. (**B**) Expression of pro-FLG, which was normalized to GAPDH. (**C**–**G**) Abundance of (**C**) FLG (34 kDa), (**D**) IVL (68 kDa), (**E**) LOR (26 kDa), (**F**) OCC (59 kDa), and (**G**) ZO-1 (187 kDa), protein levels quantified by band density and normalized to ß-actin (43 kDa). The results were expressed as means ± SD (*n* = 6). ^###^ *p* < 0.001 vs. normal (DNCB untreated group), * *p* < 0.05, ** *p* < 0.01, and *** *p* < 0.001 vs. DNCB (negative control; DNCB treated group), PD (positive control; prednisolone 3 mg/kg) treatment group, LE (LE 50, 100 or 200 mg/kg) treatment group, and ^††^ *p* < 0.01, ^†††^ *p* < 0.001 vs. PD (positive control; prednisolone 3 mg/kg) treatment group. FLG, filaggrin; LOR, loricrin; IVL, involucrin; ZO-1, zonula occludens-1; OCC, occludin; pro-FLG, pro-filaggrin.

**Table 1 nutrients-13-03209-t001:** Primer sequences (in vivo).

Gene	Forward (5′–3′)	Reverse (5′–3′)
IL-1β (m)	TGT GTT TTC CTC CTT GCC TCT GAT	TGC TGC CTA ATG TCC CCT TGA AT
IL-4 (m)	ACG GAG ATG GAT GTG CCA AAC	AGC ACC TTG GAA GCC CTA CAG A
IL-5 (m)	TCA GCT GTG TCT GGG CCA CT	TT ATG AGT AGG GAC AGG AAG CCT CA
IL-6 (m)	CCA CTT CAC AAG TCG GAG GCT TA	GCA AGT GCA TCA TCG TTG TTC ATA C
IL-12 (m)	TGA ACT GGC GTT GGA AGC	GCG GGT CTG GTT TGA TGA
IL-13 (m)	CAA TTG CAA TGC CAT CTA CAG GAC	CGA AAC AGT TGC TTT GTG TAG CTG A
IL-17 (m)	AAG GCA GCA GCG ATC ATC C	GGA ACG GTT GAG GTA GTC TGA G
IL-22 (m)	CAG CTC CTG TCA CAT CAG CGG T	AGG TCC AGT TCC CCA ATC GCC T
IL-25 (m)	CTC AAC AGC AGG GCC ACT C	GTC TGT AGG CTG ACG CAG TGT G
IL-31 (m)	ATA CAG CTG CCG TGT TTC AG	AGC CAT CTT ATC ACC CAA GAA
IL-33 (m)	GAT GAG ATG TCT CGG CTG CTT G	AGC CGT TAC GGA TAT GGT GGT C
IFN-γ (m)	CGG CAC AGT CAT TGA AAG CCT A	GGC ACC ACT AGT TGG TTG TCT TTG
TNF-α (m)	TAC TGA ACT TCG GGG TGA TTG GTC	CAG CCT TGT CCC TTG AAG AGA ACC
TSLP (m)	TGC AAG TAC TAG TAC GGA TGG GGC	GGA CTT CTT GTG CCA TTT CCT GAG
TARC (m)	TGA GGT CAC TTC AGA TGC TGC	ACC AAT CTG ATG GCC TTC TTC
MDC (m)	CAG GCA GGT CTG GGT GAA	TAA AGG TGG CGT CGT TGG
RANTES (m)	GGA GTA TTT CTA CAC CAG CAG CAA	GGC TAG GAC TAG AGC AAG CAA TGA C
CCR4 (m)	TCT ACA GCG GCA TCT TCT TCA T	CAG TAC GTG TGG TGG TGC TCT G
Pro-filaggrin (m)	GAA TCC ATA TTT ACA GCA AAG CAC CTT G	GGT ATG TCC AAT GTG ATT GCA CGA TTG
GAPDH (m)	ACT TTG TCA AGC TCA TTT CC	TGC AGC GAA CTT TAT TGA TG

## Data Availability

The data presented in this study are available on request from the corresponding author.

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
