# Peer review of "Lithospermum erythrorhizon Alleviates Atopic Dermatitis-like Skin Lesions by Restoring Immune Balance and Skin Barrier Function in 2.4-Dinitrochlorobenzene-Induced NC/Nga Mice"

_nutrients, 2021, doi:10.3390/nu13093209_

Round 1
Reviewer 1 Report
- The introduction is way too long. Get to the point more quickly.
- The ambiguous word "significantly" is over used.
- The methodology could have been stronger if a blinded rater had been used; this limitation could be mentioned in the discussion.
- I don't think we can conclude "LE 588 reduced scratching behavior by inhibiting the release of IgE and histamine in NC/Nga 589 mice" because while they may have been associated, we don't know whether the association is causal.
Author Response
늦어서 죄송합니다.
검토해 주셔서 감사합니다.
첨부파일을 봐주세요
안부

Reviewer 2 Report
“Lithospermum erythrorhizon alleviates atopic dermatitis-like 2 skin lesions by restoring immune balance and skin barrier 3 function in 2.4-dintrochlorobenzene-induced NC/Nga mice”
Overall this study found Lithospermum erythrorhizon reduced the symptoms of atopic dermatitis. Most of this appears well-done, however, there were several issues that need clarification.
- The stratum corneum is more responsible for the skin barrier than the stratum granulosum. Please clarify this throughout.
- You cannot say that “water loss, were recovered” (line 570) if you did not measure Transepidermal water loss (TEWL). The study would be stronger if you had measured TEWL.
- Specify what the “feed” was (line 152). The food you feed the mice make a big difference in your results. You need to specify the catalog number of the food, such that someone could lookup what was in the food.
- Your treatment groups are very confusing. Methods (lines 163-172)- The “normal” and “DNCB” groups appeared to be treated exactly the same. Only later in the text is the difference understandable. However, using the terms “normal” and “control” were confusing throughout. Please clarify in the methods the group that never received DNCB. Also, please use different words than “normal” and “control”. I suggest “untreated” and “DNCB only”, but other terms may also make sense. Please also clarify the purpose of the 0.5% carboxymethyl cellulose in the methods section (line 171). Was this the vehicle for the PD and LE? If yes, then specify this.
- Lines 196-197. “six locations” of what? In the histological analysis, did you take six pictures in different microscope fields per mouse, count all cells per picture, and then average them? Or did you take one picture per mouse and count cells and epidermal thickness in six locations within one image. Please clarify. Also, if you only took one picture per mouse, then you did not do enough analysis, as there can be a lot of variability within a skin sample. Line 328-329 – again this analysis is not clear in the figure legend.
- Lines 153-154, 200-203 & 210-211 are repetitive. Just state how you collected the blood once. Then describe how you made the serum once.
- Line 242: which cDNA synthesis kit did you use?
- ANOVA and Tukey HSD assume normally distributed data. What statistical test did you perform to confirm that your data was normally distributed? Also, how did you determine that six mice per group was enough? Also, line 165 states eight mice per group were assigned, but all figure legends list n=6.
- Are there statistically significant differences between the PD and LE groups? ANOVA followed by Tukey should provide this comparison. Please indicate all of these in the figures. Instead of using the # and * you could use different letters to indicate significant difference. Alternatively, you could use a different symbol for “compared to PD”. You mention in the results and discussion the dose of LE that is similar to the PD group or has higher efficacy. You cannot say this unless you show us the statistics in the figures, or provide p values in the text of the results.
- There are lots of long and sometimes run-on sentences. Please shorten. These include lines: 64-66 “TARC …; lines 290-293 “The schedule…”; 384-389 “In addition …”; 407-411 “Gene expression …; 472-476 “In addition…; 477-481 “The serum ….
Minor writing issues:
- Line 150: Spell out SLC. Abbreviations need to be spelled out at first use, and then used consistently throughout.
- Line 293, “aggravated”. All score in the DNCB only group went down with time, how is this “aggravated”?
- I find having the A-a, A-b etc for your panels in the legend confusing. Just use one letter. Also, each of the image panels should also have a letter.
- Line 517, “mediate” is the wrong word here. I suggest “resolve”.
- Lines 520-522 & 524-525 are repetitive of the text in the above paragraph. Just incorporate all of this discussion together.
- Line 567, spell out TCS.
Author Response
Forgive me for being late.
Thank you for reviewing.
Please see the attachment
Kind regards

Round 2
Reviewer 2 Report
The majority of issues were corrected in the revision. However, FLG, LOR, and IVL primarily localize to the stratum granulosum, like you had written in your original draft. My comment on the stratum corneum, is that this site is more responsible for the skin barrier. Just changing SG to SC throughout the text was not appropriate, as this is not the primary localization site for these proteins. Degradation proteins of FLG localize to the startum corneum. You may just want to call them barrier function proteins instead of their location.
Author Response
Thank you again for reviewing our manuscript.
We thank the judges for their good suggestions.
Please see the attachment.
